# Improving Protein Sequence Design through Designability Preference Optimization

## Abstract

Protein sequence design methods have demonstrated strong performance in sequence generation for *de novo* protein design. However, as the training objective was sequence recovery, it does not guarantee designability–the likelihood that a designed sequence folds into the desired structure. To bridge this gap, we redefine the training objective by steering sequence generation toward high designability. To do this, we integrate Direct Preference Optimization (DPO), using AlphaFold pLDDT scores as the preference signal, which significantly improves the in silico design success rate. To further refine sequence generation at a finer, residue-level granularity, we introduce Residue-level Designability Preference Optimization (ResiDPO), which applies residue-level structural rewards and decouples optimization across residues. This enables direct improvement in designability while preserving regions that already perform well. Using a curated dataset with residue-level annotations, we fine-tune LigandMPNN with ResiDPO to obtain EnhancedMPNN, which achieves a nearly 3-fold increase in in silico design success rate (from 6.56% to 17.57%) on a challenging enzyme design benchmark.

## 1 Introduction

The computational design of proteins with specific functions has considerable potential for solving outstanding challenges in medicine and biotechnology Huang et al. (2016); Watson et al. (2023); Vázquez Torres et al. (2025); Kim et al. (2024). Current computational protein design pipelines often decouple the problem into two stages–first generating a backbone structure, then "inverse folding" to assign a sequence predicted to adopt that backbone Watson et al. (2023); Dauparas et al. (2022). While significant progress has been made in protein sequence design (PSD) using deep learning Ingraham et al. (2019); Jing et al. (2020); Liu et al. (2022); Wang (2022); Gao et al. (2022); Anand et al. (2022); Jain et al. (2022); Zhou et al. (2023); Zheng et al. (2023); Gao et al. (2023; 2024), surpassing traditional physics-based methods like Rosetta Zaidman et al. (2020), there are remaining challenges. Existing methods primarily optimize for sequence recovery–the ability to reproduce native sequences given their backbones. However, for protein structure and function design success, what is most important is how closely the designed sequence folds to the designed target structure Ruffolo et al. (2025); Zhu et al. (2025); Johansen et al. (2025); Hou et al. (2025). As expected, there is a strong correlation between designability and experimental success for both protein binder design Bennett et al. (2023) and enzyme design Kim et al. (2024). A key challenge is that state-of-the-art PSD models, while optimized for sequence recovery, often exhibit poor designability Ye et al. (2024). These models can introduce subtle yet critical errors, such as placing structure-breaking residues like Proline and Glycine within a desired $\beta$-sheet, causing it to misfold into a flexible loop (Fig. 1). This low reliability necessitates generating thousands of candidates to find a few viable ones, leading to success rates as low as 3% for enzyme design with standard RFDiffusion and ProteinMPNN pipelines Watson et al. (2023). This brute-force screening is computationally inefficient, slows experimental validation, and highlights an urgent need for methods that directly optimize for designability.

We set out to directly optimize a ProteinMPNN-like model inspired by the success of alignment techniques like Reinforcement Learning from Human Feedback (RLHF) Christiano et al. (2017) used to bridge objective gaps in large language models (LLMs) Lla; DeepSeek-AI (2025). We aimed to explicitly align PSD models towards generating sequences with high designability. Proteins offer advantages over natural language for this adaptation. First, instead of relying on subjective human preferences, we can leverage physics-based simulation Wang et al. (2024) or high-accuracy structure

Figure 1: ResiDPO directly optimizes protein designability by decoupling optimization at the residue level. While standard models often generate sequences with local structural flaws (low pLDDT), ResiDPO selectively applies a preference reward to improve these problematic regions (such as Proline and Glycine in the figure which break the helix). To avoid catastrophic forgetting, it simultaneously preserves high-confidence regions using KL regularization. This decoupled strategy boosts the *in silico* design success rate from 6.56% to 17.57%.

predictors Senior et al. (2020); Baek et al. (2021). We chose the predicted Local Distance Difference Test (pLDDT) score from AlphaFold2 Jumper et al. (2021), which correlates well with structural accuracy (Fig. 6) and provides a quantitative, objective reward signal for designability. Second, the fixed length of sequences designed for a specific backbone allows for fine-grained, residue-level reward assignment, unlike the sequence-level rewards typically used in LLMs.

Building on these insights, here we introduce Residue-level Designability Preference Optimization (ResiDPO), an adaptation of the Direct Preference Optimization (DPO) Rafailov et al. (2023) framework tailored for protein design. Standard DPO, while effective for aligning language models using sequence-level preferences, faces challenges when applied naively to PSD. It optimizes a single loss function balancing preference learning against a KL-divergence term that regularizes towards the original model distribution. This can create conflicting gradients, especially when trying to significantly increase the probability of high-designability sequences. ResiDPO overcomes this by leveraging residue-level pLDDT scores as rewards. It *decouples* the DPO loss: for residues predicted to improve designability (e.g., low initial pLDDT), it prioritizes maximizing the preference reward signal; for residues already contributing positively to the structure (e.g., high pLDDT and high confidence from the base model), it prioritizes KL regularization to maintain learned structural features. This decoupling provides a clearer, more stable optimization target, directly enhancing designability without catastrophic forgetting (Fig. 1). We use ResiDPO to develop an improved model, EnhancedMPNN, which increases the design success rate nearly 3-fold on a range of challenging enzyme and binder design challenges.

Our contributions are summarized as follows:

- We frame the challenge of low designability in PSD as a critical objective misalignment problem, amenable to targeted alignment strategies inspired by reinforcement learning.
- We propose ResiDPO, a novel alignment algorithm that adapts and extends DPO for protein design by leveraging objective, residue-level structural feedback (pLDDT) to decouple the optimization objective, enabling more effective learning of designability.
- We curate and will release a large-scale dataset featuring residue-level pLDDT labels, providing a valuable resource for training and evaluating designability-focused PSD models.
- We demonstrate empirically that our ResiDPO-finetuned model, EnhancedMPNN, achieves substantial improvements in in silico design success rates (nearly 3x for enzymes, 2x for binders), significantly reducing computational cost and accelerating the design-build-test cycle for functional proteins.

## 2  RELATED WORK

**Protein Sequence Design (Inverse Folding).** Protein sequence design methods aim to generate amino acid sequences that fold into a desired 3D structure. These methods have seen significant progress, driven by advancements in graph neural networks and attention mechanisms. Early work like GraphTrans Ingraham et al. (2019) extended the Transformer architecture Vaswani et al. (2017)

to capture long-range interactions in protein sequences by modeling sparse relationships between residues distant in sequence but proximal in 3D space. Geometric Vector Perceptrons (GVPs) Jing et al. (2020) further enhanced it by incorporating geometric information, representing residues and their relationships as vectors and scalars. This approach was adopted by ESM-IF1 Hsu et al. (2022), which combined a GVP-based structure encoder with a GVP-Transformer to learn inter-residue interactions. GCA Tan et al. (2023) focused on incorporating global context. ProteinMPNN Dauparas et al. (2022) extends GraphTrans by introducing more geometry features and random decoding. PiFold Gao et al. (2022) introduced a novel residue featurization, a PiGNN module, and further improved efficiency by dispensing with autoregressive decoding. More recent efforts have explored knowledge integration (KW-Design Gao et al. (2023)) and adaptation of pre-trained language models (LM-Design Zheng et al. (2023), ZymCTRL Munsamy et al. (2024), CarbonDesign Ren et al. (2024)) for sequence design. These methods have collectively improved natural sequence recovery rates from approximately 40% to over 60%.

However, maximizing natural sequence recovery does not directly address the crucial aspect of designability—the ability of a designed sequence to fold into the target structure. As highlighted by Huang et al. Huang et al. (2016), the space of natural sequences represents only a small fraction of possible sequences, and multiple sequences can fold into the same structure. Therefore, ensuring high designability is paramount. Unlike existing methods that primarily focus on recovering natural sequences, this work explicitly aims to improve the designability of protein sequence design methods.

Many proteins, particularly enzymes, do not function in isolation. They often require cofactors, such as ligands and metals, for activity. LigandMPNN Dauparas et al. (2025) extends ProteinMPNN to incorporate ligand awareness. Given the challenges inherent in *de novo* enzyme design, we adopt LigandMPNN as our base model.

**Preference learning.** Self-supervised language models have demonstrated promising capability in completing zero-shot tasks. Before applying them to downstream tasks, aligning with human preferences has proven crucial for improving performance on downstream tasks. Early approaches employed reinforcement learning from human feedback (RLHF) Christiano et al. (2017) to fine-tune models based on relative judgments of response quality. However, RLHF can be complex and challenging to implement. Direct Preference Optimization (DPO) Rafailov et al. (2023) significantly simplified this process by eliminating the need for an explicit reward model and instead directly optimizing a pairwise logistic loss based on preferences. Subsequent work has extended DPO by removing the Bradley-Terry assumption (IPO Azar et al. (2024)), enabling listwise optimization (LiPO Liu et al. (2024)), removing the requirement for a reference model (SimPO Meng et al. (2024)), and so on. Recently, DPO has been applied to protein sequence design. Park et al. Park et al. (2024) used DPO for peptide design, demonstrating improvements of 8% in structural similarity and 20% in sequence diversity. In this work, we focus on improving DPO to improve the designability of the designed sequences.

## 3 METHODS

This section details Residue-level Designability Preference Optimization (ResiDPO), a novel training framework specifically designed to align protein sequence design (PSD) models with the objective of *designability*. We first contextualize PSD within the preference optimization paradigm, adapting concepts from Direct Preference Optimization (DPO) used in large language models (LLMs). We then describe our strategy for generating preference data leveraging quantitative protein quality metrics. Finally, we introduce the ResiDPO objective function, which overcomes limitations of standard DPO by decoupling preference learning and model regularization at the residue level.

### 3.1 PROTEIN DESIGN AS PREFERENCE OPTIMIZATION

The goal of protein sequence design is to generate an amino acid sequence $y = (y_1, ..., y_L)$ that folds into a target backbone structure $x$. While conventional PSD models are trained via sequence recovery (maximizing $P(y_{native}|x)$), this objective is often misaligned with the desired outcome: generating novel sequences $y$ that fold precisely to the structure $x$ (designability). Here, we frame PSD as a preference optimization problem, aiming to increase the probability of sequences with high designability.

Drawing inspiration from alignment techniques in LLMs Christiano et al. (2017), we first adapt the Direct Preference Optimization (DPO) framework Rafailov et al. (2023). DPO optimizes a policy model $\pi_\theta$ to better satisfy preferences compared to a reference model $\pi_{ref}$ (typically the initial pre-trained model), using a dataset $\mathcal{D}$ of preference pairs $(y_w, y_l)$ for a given prompt $x$, where $y_w$ is preferred over $y_l$. The DPO objective maximizes the likelihood of preferred responses while regularizing against large deviations from the reference model via a KL divergence penalty:

$$\mathcal{L}_{DPO}(\pi_\theta; \pi_{ref}) = -\mathbb{E}_{(x,y_w,y_l)\sim\mathcal{D}} \left[ \log \sigma \left( \beta \log \frac{\pi_\theta(y_w|x)}{\pi_{ref}(y_w|x)} - \beta \log \frac{\pi_\theta(y_l|x)}{\pi_{ref}(y_l|x)} \right) \right], \quad (1)$$

where $\sigma$ is the sigmoid function, and $\beta$ is a hyperparameter controlling the strength of the preference relative to the regularization. In our context, $x$ is the target protein backbone structure, $y_w$ and $y_l$ are candidate amino acid sequences, $\pi_{ref}$ is the pre-trained PSD model (e.g., LigandMPNN), and $\pi_\theta$ is the model being fine-tuned.

For a quantitative measure of designability, we utilize the predicted Local Distance Difference Test (pLDDT) score derived from AlphaFold2 (AF2) Jumper et al. (2021). The pLDDT score serves as a proxy for folding accuracy and stability, correlating well with experimental success and metrics like $C\alpha$RMSD (Fig. 6). Thus, a sequence $y_w$ with a higher pLDDT score is considered preferable to a sequence $y_l$ with a lower score for the same target structure $x$.

### 3.2 PREFERENCE PAIR GENERATION

Effective DPO training relies on high-quality preference pairs $(y_w, y_l)$. While LLM alignment often requires a separate reward model or human labeling to score outputs, PSD benefits from objective quality metrics like pLDDT. To generate preference pairs for a given structure $x$, we first sample multiple candidate sequences using the reference model $\pi_{ref}$. We then predict the structure and pLDDT score for each sampled sequence using AF2. Based on pLDDT scores, we explored different strategies for selecting $(y_w, y_l)$ pairs:

**Rejection Sampling**: This approach is analogous to methods used in the post-training alignment of LLMs Grattafiori et al. (2024); DeepSeek-AI (2025), where outputs are scored by a reward model. In our context, for a given structure $x$, we identify the sequence with the highest pLDDT score in the sampled batch as the preferred sequence $y_w$. Other sequences from the same batch, with lower pLDDT scores, are then randomly selected to serve as less preferred sequences $y_l$, forming multiple pairs $(y_w, y_l)$. We adopt this method as a baseline for comparison.

**Application Sampling**: Inspired by practical protein design workflows where sequences below a certain quality threshold are discarded Watson et al. (2023); Kim et al. (2024), this method selects $y_w$ as a sequence with pLDDT ¿ 80 and $y_l$ as a sequence with pLDDT ¡ 75. However, the scarcity of high-pLDDT generations (pLDDT ¿ 80) for many structures using current PSD models severely limits the number of preference pairs obtainable with this approach (Fig. 7), reducing dataset diversity.

**Relative Sampling**: To generate more preference pairs and better utilize the available data, this method selects any pair of sequences $(y_i, y_j)$ generated for the same structure $x$ such that their pLDDT scores differ by more than a predefined threshold $\delta$: $pLDDT(y_i) - pLDDT(y_j) > \delta$. The sequence with the higher pLDDT is designated $y_w$, and the one with the lower score is $y_l$.

Ablation studies indicated that Relative Sampling with a pLDDT difference threshold $\delta = 10$ yielded the best performance in downstream tasks. Therefore, we adopted this strategy for generating the preference dataset $\mathcal{D}$ used in subsequent experiments.

### 3.3 RESIDUE-LEVEL DESIGNABILITY PREFERENCE OPTIMIZATION (RESIDPO)

The standard DPO objective (Eq. 1) optimizes a single loss, implicitly balancing preference maximization and regularization (KL divergence from $\pi_{ref}$). In PSD, optimizing the entire sequence probability to favor $y_w$ can conflict with the KL constraint, particularly when substantial sequence changes are needed for improved designability. Modifying probabilities at certain residues to improve predicted local structure (higher pLDDT) inevitably increases the KL divergence, potentially hindering optimization or leading to instability.

Unlike variable-length text generation, protein design for a fixed backbone structure yields fixed-length sequences ($L$). This enables a residue-level analysis of designability via per-residue pLDDT scores, $pLDDT(y, i)$ for residue $i$ in sequence $y$. ResiDPO leverages this granularity to decouple the optimization objectives. Instead of a single sequence-level objective, it applies targeted updates at the residue level, separating preference learning from constraint enforcement.

ResiDPO decomposes the loss into two components: Residue-level Preference Learning (RPL) and Residue-level Constraint Learning (RCL).

### 3.3.1 RESIDUE-LEVEL PREFERENCE LEARNING (RPL)

RPL focuses the preference optimization on specific residue positions where the preferred sequence $y_w$ shows significant improvement in local predicted structure compared to the less preferred sequence $y_l$. The RPL loss is defined as:

$$\mathcal{L}_{\text{RPL}} = -\mathbb{E}_{(x, y_w, y_l) \sim \mathcal{D}} \left[ \log \sigma \sum_{i \in \mathcal{I}} \left( \frac{\log \pi_\theta(y_w^i | x) - \log \pi_\theta(y_l^i | x)}{|\mathcal{I}|} \right) \right], \quad (2)$$

where $\mathcal{I}$ is the set of residue indices $i$ where the pLDDT difference exceeds a margin $\alpha$:

$$\mathcal{I} = \{i \,|\, pLDDT(y_w, i) - pLDDT(y_l, i) > \alpha\}. \quad (3)$$

Here, $y_k, i$ denotes the amino acid at position $i$ of sequence $y_k$. This loss term encourages the model $\pi_\theta$ to increase the relative log-probability of preferred residues, specifically at positions deemed important for improving designability.

If no residues satisfy the condition ($\mathcal{I} = \phi$), indicating similar local pLDDT profiles or global differences, we revert to a sequence-level preference signal by setting $\mathcal{I} = \{1, ..., L\}$ for that pair, effectively applying the standard DPO logic over the entire sequence.

### 3.3.2 RESIDUE-LEVEL CONSTRAINT LEARNING (RCL)

RCL aims to preserve the knowledge of the reference model $\pi_{ref}$ at positions that are already well-structured and confidently predicted, preventing catastrophic forgetting and maintaining structural integrity. It applies a KL-divergence penalty at specific residue positions:

$$\mathcal{L}_{\text{RCL}} = \mathbb{E}_{(x, y_w, y_l) \sim \mathcal{D}} \left[ \sum_{j \in \mathcal{J}} \frac{\pi_{ref}(y_w^j | x) \cdot \log \frac{\pi_{ref}(y_w^j | x)}{\pi_\theta(y_w^j | x)}}{|\mathcal{J}|} \right], \quad (4)$$

where

$$\mathcal{J} = \{j \,|\, pLDDT(y_w, j) > \beta \cap \pi_{ref}(y_w, j | x) > \gamma\} \quad (5)$$

is the set of residue positions $j$ where the pLDDT is above a threshold $\beta$ and the reference model's probability exceeds a confidence threshold $\gamma$. The RCL loss encourages $\pi_\theta$ to stay close to $\pi_{ref}$ at these "high-quality, high-confidence" positions $\mathcal{J}$.

### 3.3.3 COMBINED RESIDPO LOSS

The final ResiDPO loss function combines the residue-level preference learning and constraint learning terms:

$$\mathcal{L}_{\text{ResiDPO}} = \mathcal{L}_{\text{RPL}} + \lambda \mathcal{L}_{\text{RCL}}, \quad (6)$$

where $\lambda$ is a hyperparameter balancing the strength of the preference signal against the constraint encouraging preservation of the reference model's reliable predictions. This decoupled, residue-level optimization allows ResiDPO to more effectively navigate the complex trade-off between improving designability (via RPL) and maintaining model stability and learned structural features (via RCL), leading to improved performance in generating designable protein sequences.

# 4 EXPERIMENTS

In this section, we present the results of our proposed ResiDPO for protein sequence design. We begin by describing the construction of our training and evaluation dataset, followed by an ablation study of key hyperparameters. Then we demonstrate the practical impact of ResiDPO on a challenging enzyme design benchmark. Finally, we present a detailed analysis of ResiDPO's performance compared to standard DPO.

## 4.1 PDB-D DATASET AND EVALUATION METRICS

To train and evaluate our models, we curated PDB-D, a diverse and high-quality dataset of protein structures with corresponding per-residue pLDDT labels using AlphaFold2 (AF2) Jumper et al. (2021). To ensure the data quality, we curated monomeric structures from the Protein Data Bank (PDB) Berman et al. (2000) determined by X-ray crystallography with a resolution better than 3.5 Å. We restricted our dataset to proteins shorter than 1,000 residues for three principal reasons: (1) AF2 structure prediction for larger proteins is computationally demanding; (2) current backbone generation models often encounter challenges in designing very large structures; and (3) shorter protein sequences are frequently preferred in practical applications. To prevent data leakage, we adopted a rigorous data-splitting strategy. Structures released after September 30, 2021, were designated as the validation set, consistent with prior work Abramson et al. (2024). We performed structure-based clustering and treated clusters containing any validation structure as validation clusters. All structures in the remaining clusters formed the training set. This process resulted in a training set of 19,203 structures and a validation set of 1,690 representative structures. For each structure, we used LigandMPNN Dauparas et al. (2025) to generate eight sequences at a temperature of 1.0 to encourage diversity. These sequences were then predicted by AF2 to obtain per-residue pLDDT scores (Fig. 1).

As described above, to evaluate the designability of sequences on design benchmarks, we employ the gold standard: running full AF2 predictions. We evaluated design success based on the quality of the predicted AF2-structures and pLDDT scores. However, this approach incurs significant computational cost. To enable efficient iteration and evaluation during our extensive ablation studies, we propose the new metric **pLDDT Accuracy**. Unlike the full AF2 evaluation used for the final benchmark, pLDDT Accuracy is an efficient proxy that measures the correlation between the model's output likelihood for a validation sequence and the sequence's actual pLDDT score. This allows us to effectively evaluate the model's success in learning to generate sequences likely to be designable (i.e., achieve high pLDDT) without the computational burden of running AF2 for every ablation configuration. We also track sequence recovery on the LigandMPNN validation set to understand how far the model moves from the original output distribution.

## 4.2 EXPERIMENTAL DETAILS

Due to the importance of ligand context in enzyme design, we employed LigandMPNN Dauparas et al. (2025), a ligand-aware variant of ProteinMPNN, as our base model. We first conducted a comprehensive grid search to optimize hyperparameters for standard DPO. Subsequently, we utilized these optimized hyperparameters in our proposed ResiDPO method. Unless otherwise stated, all training runs utilized the Adam Kingma & Ba (2015) optimizer with a learning rate of 5e-7 for 100,000 iterations. We implemented a learning rate schedule consisting of a 3% warmup period followed by cosine annealing decay. The training was distributed across two L40 GPUs, utilizing a total batch size of 8 with a gradient accumulation factor of 16.

For ResiDPO, we employed the following parameter settings: $\alpha = 10$, $\beta = 80$, $\gamma = 0.5$, and $\lambda = 0.01$. We set $\beta$ to 80 based on the observation that, in most practical protein design applications, a pLDDT score greater than 80 generally indicates sufficient designability. Since $\beta$ and $\gamma$ are highly correlated, this choice helps reduce the search space, allowing us to only look for the best value for $\gamma$.

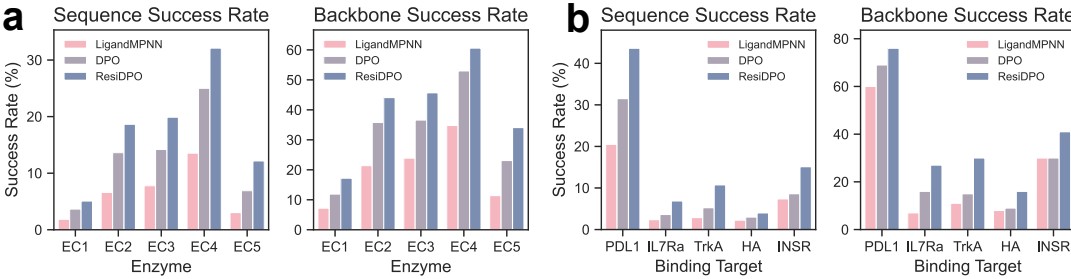

Figure 2: Design success rates for LigandMPNN, DPO-finetuned LigandMPNN, and ResiDPO-finetuned LigandMPNN (EnhancedMPNN) on the enzyme design (a) and binder design (b) benchmarks. EnhancedMPNN, trained with our ResiDPO method to directly optimize for designability, demonstrates significantly higher design success rates. This enhanced designability allows for the successful sequence design of traditionally "undesignable" backbones. The consistent improvement observed in the binder design benchmark highlights the generality of the ResiDPO approach for improving designability across diverse protein design problems, including protein-protein interaction.

## 4.3 DESIGN BENCHMARK RESULTS

### 4.3.1 ENZYME DESIGN BENCHMARK

To evaluate the performance of ResiDPO, we used the enzyme active site scaffolding benchmark from RFDiffusion2 Ahern et al. (2025). This benchmark provides five enzymes from five EC classes with defined catalytic pockets (We expanded the benchmark into 41 enzymes in Appendix Sec. A.7). For each enzyme, we generated 1,000 backbones based on the catalytic pocket (Tab. 3) using RFDiffusion2 and designed eight sequences per backbone using LigandMPNN, DPO-finetuned LigandMPNN, and ResiDPO-finetuned LigandMPNN (EnhancedMPNN) at a temperature of 0.1. The catalytic residues were fixed (undesigned) throughout both backbone and sequence generation processes. We assessed design success based on the criteria of pLDDT 80 and C$\alpha$ RMSD 1.5 Å.

As illustrated in Fig. 2a, ResiDPO significantly improved the sequence design success rate compared to both LigandMPNN and DPO. EnhancedMPNN achieved an average sequence design success rate of 17.57%, a nearly threefold improvement over ligandMPNN (6.56%). This improvement was consistent across all five enzymes. Furthermore, the increased designability also led to a significant increase in backbone success rate (i.e., fraction of backbones with at least one successful sequence), from 19.74% for LigandMPNN to 40.34% for EnhancedMPNN. This indicates that ResiDPO *not only improves sequence design but also expands the set of designable backbones*, potentially recalls more designable *de novo* proteins, and saves significant computational resources.

### 4.3.2 BINDER DESIGN BENCHMARK

To assess the generalization capabilities of our approach beyond the monomeric proteins prevalent in the PDB-D training dataset, we evaluated ResiDPO on a challenging protein binder design task. We utilized the binder benchmark set introduced by RFdiffusion Watson et al. (2023), for which models must design sequences that promote specific inter-chain interactions. For each target protein in the benchmark, 100 backbone structures were generated using RFdiffusion. Subsequently, for each backbone, the baseline LigandMPNN, the DPO-finetuned LigandMPNN, and the ResiDPO-finetuned LigandMPNN were tasked with designing eight sequences. Following RFdiffusion, the in silico success of a designed sequence was defined by three criteria: inter-chain PAE ¡ 10, C$\alpha$ RMSD ¡ 1 Å, and pLDDT ¿ 80. As shown in Fig. 2b, the baseline LigandMPNN achieved a sequence design success rate of 7.07%. Fine-tuning with standard DPO modestly increased this rate to 10.40%. EnhancedMPNN, leveraging ResiDPO, achieved a sequence design success rate of 16.07%. This represents a substantial improvement, more than doubling (approximately 2.27-fold) the success rate of the baseline LigandMPNN. These results demonstrate that ResiDPO, despite not being explicitly trained on protein complexes or interaction data, confers a strong generalization for improving designability in complex multichain systems.

Table 1: Comparison of different sampling methods and loss components of ResiDPO on the validation set. The best results of each section are labeled in bold.

| Sampling | Method | Seq. Recovery | pLDDT Acc. |
|---|---|---|---|
| - | LigandMPNN | 57.63 | 57.71 |
| Rejection Sampling | DPO | 56.86 | 61.23 |
| Application Sampling | DPO | 56.29 | 61.67 |
| Relative Sampling | DPO | **57.03** | **62.11** |
| Relative Sampling | RPL | 54.23 | 63.44 |
| Relative Sampling | ResiDPO | **55.56** | **66.08** |

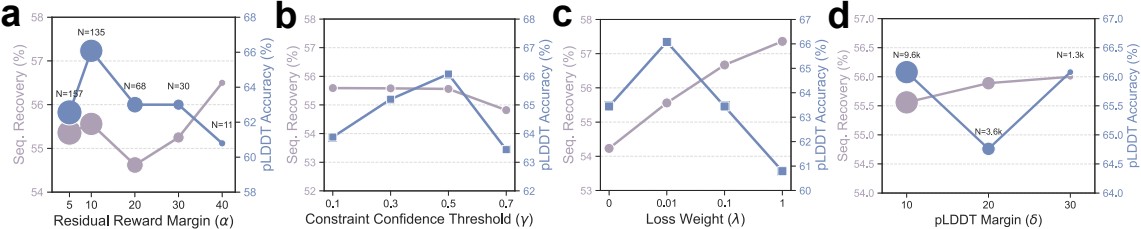

Figure 3: Ablations on the hyperparameters of ResiDPO. The marker size N in panels a and d indicates the number of selected residue pairs ($|\mathcal{I}|$) and remaining backbones, separately.

## 4.4 ABLATION STUDY

Our ablation studies validate the design of ResiDPO by systematically evaluating its core components and hyperparameter sensitivities. We first established that Relative Sampling is the optimal strategy for generating preference pairs for the DPO baseline, yielding a pLDDT accuracy of 62.11% (Tab. 1). Isolating our Residue-level Preference Learning (RPL) component improved pLDDT to 63.44% but at the cost of sequence recovery (54.23%). The full ResiDPO algorithm, which integrates RPL with Residue-level Constraint Learning (RCL), achieved the highest pLDDT accuracy of 66.08% while maintaining strong sequence recovery (55.56%). This demonstrates that RCL is crucial for preventing knowledge degradation from the pretrained model, synergizing with RPL to effectively learn preferences without catastrophic forgetting.

Key hyperparameters were meticulously tuned for ResiDPO to balance performance and robustness (Fig. 3). The PRL margin ($\alpha = 10$), RCL confidence threshold ($\gamma = 0.5$), and loss weight ($\lambda = 0.01$) were identified to maximize pLDDT accuracy with decent sequence recovery. For preference pair construction, a pLDDT margin of $\delta = 10$ was chosen over $\delta = 30$; while both yielded similar top pLDDT scores, $\delta = 10$ offered a significantly larger and more diverse training set (9,557 vs. 1,283 pairs), promoting better generalization.

## 4.5 WHAT MAKES IT DIFFERENT: HOW RESIDPO ENHANCES PROTEIN DESIGNABILITY?

To understand the mechanisms by which ResiDPO enhances designability, we analyzed the changes in amino acid composition induced by EnhancedMPNN on both natural (validation set from PDB-D) and designed (enzyme benchmark) backbones.

We generated confusion matrices (Fig. 10) to visualize the residue substitutions introduced by EnhancedMPNN. A consistent trend emerged across both natural and designed backbones: EnhancedMPNN tended to introduce more charged residues, particularly Glutamic Acid (E), Lysine (K), and Arginine (R). Almost 50% of the original amino acids that were originally Alanine (A), Glutamine (Q), Serine (S), and Threonine (T) are mutated to others by EnhancedMPNN. Proline (P) and Glycine (G) remained largely unchanged.

We also compare the overall amino acid distributions generated by different methods for both types of backbones (Fig. 4). The observed trends aligned with the residue-level analysis. Notably, Alanine (A) abundance is significantly reduced in designed backbones, particularly with ResiDPO. While DPO and ResiDPO generally exhibit similar trends, ResiDPO often induces more pronounced changes. For instance, both methods tend to decrease the frequency of Alanine (A) and increase the frequency of Glutamic acid (E) in both natural and designed backbones, but ResiDPO's effect is considerably stronger.

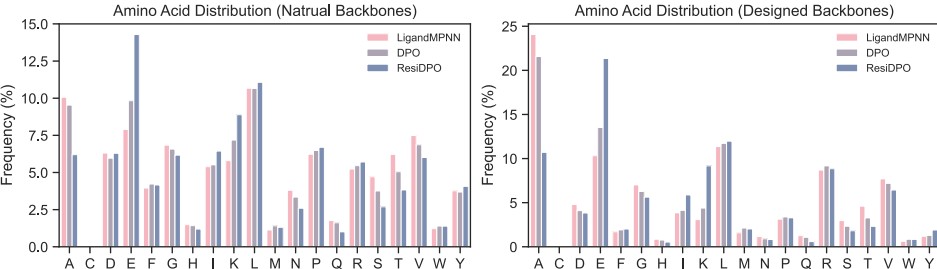

Figure 4: Comparison of amino acid distributions generated by different methods on natural backbones (left) and designed backbones (right).

The overall sequence distribution changes can be understood by considering how amino acid sequences determine protein 3D structures. To a first approximation, polar and charged residues are on the protein surface, where they can interact with water, while hydrophobic residues are buried away from water in the protein core. Structure prediction methods such as AF2, learn these (and many other) patterns. Alanine residues can be buried or exposed, and hence are somewhat ambiguous, as are the small polar residues serine and threonine. In contrast, glycine and proline have unique conformational preferences, making them relatively unambiguous (glycine can adopt conformations inaccessible to other residues, while proline is limited to a subset of local conformations). Hence the EnhancedMPNN overall can be viewed as reducing the ambiguity in the sequence-structure mapping, which increases the AF2 predicted confidence.

### 4.6 How many backbones are needed for effective optimization?

Beyond enhancing sequence designability, ResiDPO offers promise for optimizing other functional properties, particularly those constrained by the high cost of wet-lab experiments and data acquisition. We assessed its data efficiency by comparing ResiDPO against DPO using varying numbers of randomly selected training backbones (Fig. 5).

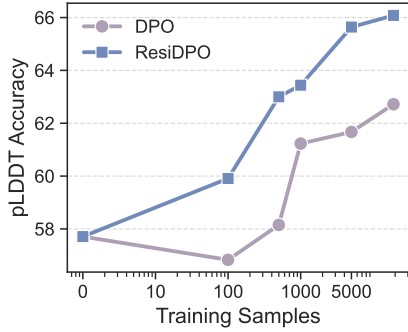

ResiDPO demonstrated a clear advantage, particularly in low-data regimes. With only 100 samples, DPO's pLDDT accuracy declined, while ResiDPO, leveraging its targeted residual-level supervision, learned effectively. Notably, ResiDPO with just 1k samples achieved comparable pLDDT accuracy (63.44%) to DPO with 19k samples—a significant data efficiency gain. Given that only 502 of the randomly sampled backbones met our selection criterion ($\delta = 10$), our findings suggest that ResiDPO can

Figure 5: ResiDPO achieves good performance with only a thousand training samples.

achieve substantial performance improvements with a modest set of ~**500** carefully chosen backbones, highlighting its practical applicability for broader, resource-constrained optimization tasks.

## 5 Conclusion

In this work, we introduced Residue-level Designability Preference Optimization (ResiDPO), a novel training strategy designed to bridge the gap between training sequence recovery and inference designability. By performing per-residue comparison, we effectively decouple the traditionally conflicting objectives of Direct Preference Optimization (DPO) into two distinct components: reward learning from improved positions and constraint learning for already well-predicted positions. This refined approach provides a more targeted and effective optimization process, leading to a nearly three-fold increase in design success for enzymes and a two-fold improvement for protein binders. What's more, by making previously undesignable backbones designable, ResiDPO meaningfully expands the designable space for both backbones and sequences, potentially prompting advancements in backbone design methodologies. While we have demonstrated the efficacy of ResiDPO using LigandMPNN, the method is architecture-agnostic and readily generalizable to other sequence design models. ResiDPO can also be readily generalized to a wide range of design tasks to improve properties such as stability and expressibility. Experimental characterization of EnhancedMPNN-generated sequences will reveal whether the enhanced in silico foldability translates to greater control over structure and higher design success rates.

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
