# OpenReview forum: "Improving Protein Sequence Design through Designability Preference Optimization"
_ICLR.cc/2026/Conference — ICLR 2026 Conference Withdrawn Submission_

### Official Review · Reviewer_eyxe · 2025-10-30

**Soundness:** 3
**Presentation:** 3
**Contribution:** 2
**Rating:** 4
**Confidence:** 4

**Summary:**

The paper applies DPO to protein sequence design by using residue-level pLDDT scores from AlphaFold2 as rewards. They split the loss into two parts - one optimizes low-pLDDT residues (preference learning), one preserves high-pLDDT residues (KL regularization). This gives "EnhancedMPNN" which improves enzyme design success from 6.56% to 17.57%.

**Strengths:**

The problem is well-motivated - sequence recovery doesn't equal designability, and framing this as an alignment problem makes sense.

The threefold improvement on enzyme benchmarks looks good (Fig 2a). The method also generalizes decently to binder design.

Decoupling the DPO loss at residue level is intuitive and seems to help based on the ablations.

The dataset with residue-level pLDDT labels could be useful.

**Weaknesses:**

The core contribution is pretty incremental. It's basically DPO with residue-level splitting instead of sequence-level. Section 3.3 makes it sound complicated but the idea is simple if I got it right: apply preference loss where pLDDT is low, apply KL where it's high. This isn't a major conceptual advance, more of a good engineering. The paper oversells it as "novel alignment algorithm."

Circular evaluation. You train using AF2 pLDDT and evaluate using AF2 predictions. How do we know this actually improves real designability vs just learning to game AlphaFold's confidence? Sure, pLDDT correlates with structure accuracy, but the correlation isn't perfect. Would be nice to see at least some validation on structures not from PDB - like actually designed proteins with known experimental outcomes.

Preference pair generation seems questionable. "Relative sampling" creates pairs from any sequences with pLDDT difference >10, even if both are mediocre (like 50 vs 60). The model is learning preferences from comparing bad sequences to slightly-less-bad sequences. The "application sampling" approach (high vs low quality) makes more sense but they reject it due to insufficient data. Quality-quantity tradeoff that isn't well justified.

Too many hyperparameters. The authors claim "comprehensive grid search" but don't show the full search space or give clear guidance for new applications. For someone wanting to use this on a different protein design task, what values should they use? The method seems tuned specifically to their benchmarks.

Modest improvements over standard DPO. Table 1 shows 66.08% vs 62.11% pLDDT accuracy - that's a 4 percentage point improvement. Is all the added complexity worth it? No error bars or significance tests provided. And standard DPO already gives a decent boost over baseline (57.71% to 62.11%), so DPO itself works pretty well.

The mechanism isn't well understood. Section 4.5 shows EnhancedMPNN uses more charged residues and less Alanine. The explanation about "reducing ambiguity" is vague. Why do these specific changes improve designability? Is this a general principle or an artifact of the training data? Without further insight it's hard to know if this will generalize to other design problems.

Some results look cherry-picked. Main text shows 5 enzymes, but they mention "expanding to 41 enzymes" buried in the appendix. Why not show all 41 in the main results? Makes you wonder why?

**Questions:**

How does this compare to other recent inverse folding methods like those in ProteinBench (https://proteinbench.github.io)?

The success rate is still only 17.57% - what about the 82% that fail? Any analysis of failure modes?

With α=10 for pLDDT margin, how sensitive is this?

Can you show the full hyperparameter grid search results?

---

### Official Review · Reviewer_Tc3D · 2025-10-31

**Soundness:** 2
**Presentation:** 2
**Contribution:** 2
**Rating:** 2
**Confidence:** 3

**Summary:**

This paper proposes ResiDPO, a residue-level variant of Direct Preference Optimization (DPO) for protein sequence design. The goal is to improve designability rather than sequence recovery. Using AlphaFold2 pLDDT as a quantitative preference signal, the authors decompose DPO into residue-level preference and constraint terms. Fine-tuning LigandMPNN with this method (EnhancedMPNN) leads to a 3× increase in in-silico success rate on enzyme benchmarks.

**Strengths:**

1.	The residue-level optimization idea is well-motivated and fits naturally with protein structure design, where local regions can be evaluated independently and conserved regions should remain stable.
2.	The method is conceptually sound and mathematically well-formulated, with clear derivations and intuitive design choices.

**Weaknesses:**

1.	The evaluation is limited and entirely in silico. All preference signals and success metrics rely on AlphaFold2 (AF2), raising concerns that the model may simply exploit AF2’s scoring patterns rather than genuinely improving folding or stability. No cross-validation with other structure predictors (e.g., ESMFold, RoseTTAFold) or experimental validation is provided.
2.	The set of baselines is rather limited, lacking comparisons with other sequence design methods such as ESM-IF, PiFold, or KW-Design. Including results on standard protein design benchmarks (e.g., CATH) would better demonstrate the generality of the proposed method.
3.	Some references and formatting look unpolished (e.g., wrong citation format, wrong bolding in Table 1), which hurts readability.

**Questions:**

1.	As noted in Weakness 1, could you verify whether ResiDPO-trained models generalize to structure predictors beyond AF2, such as ESMFold or RoseTTAFold?
2.	What is the computational cost of generating preference pairs with AF2, and how feasible is this pipeline for larger datasets?
3.	How well does ResiDPO transfer to other architectures (e.g., ProteinMPNN, PiFold, LM-Design) on general protein design benchmarks?

---

### Official Review · Reviewer_3uHx · 2025-11-10

**Soundness:** 3
**Presentation:** 2
**Contribution:** 2
**Rating:** 4
**Confidence:** 3

**Summary:**

This paper proposes ResiDPO (Residue-level Direct Preference Optimization), a fine-tuning framework for protein sequence design that aims to align model objectives with designability rather than sequence recovery. The method uses AlphaFold2 pLDDT scores as a proxy for foldability and constructs residue-level preference pairs to encourage the model to favor sequences that are more likely to fold into the target backbone. The resulting model, called EnhancedMPNN, is evaluated on enzyme and binder benchmarks, showing an improvement in in silico design success rate compared with LigandMPNN and standard DPO fine-tuning.

**Strengths:**

1. The misalignment between sequence-recovery training and true designability is well motivated and clearly articulated.
2. Introducing residue-level decomposition (RPL + RCL) is a technically neat way to balance preference learning and knowledge retention, avoiding catastrophic forgetting.
3. Experiments on enzyme and binder benchmarks show consistent in silico improvements in design success rate and reasonable ablations.

**Weaknesses:**

1. ResiDPO is explicitly trained to generate sequences that yield higher AlphaFold2 pLDDT scores and is then evaluated using the same metric, creating a self-consistency bias that may inflate the reported gains in design success.
Moreover, pLDDT measures local confidence within AlphaFold2 rather than true physical or thermodynamic stability.
Using it as the sole optimization constraint is therefore too weak to capture real designability and may encourage the model to exploit AlphaFold2’s scoring patterns instead of learning generalizable folding principles.
Validation with alternative predictors (e.g., ESMFold, RoseTTAFold) or complementary metrics such as PAE, energy-based stability would provide stronger evidence of genuine foldability improvement.

2. Although ResiDPO is presented as improving “designability,” the paper offers little analysis of what changes in the designed sequences drive improvement. For example, which residue types or structural motifs are favored after training, and do these align with known biophysical intuitions?   Although EnhancedMPNN shows shifts in amino-acid usage, the paper does not analyze whether these changes correspond to physically meaningful improvements or merely exploit AF2 preferences.

3. The benchmarks (enzymes and binders) are both relatively small and structurally homogeneous. It remains unclear whether ResiDPO generalizes to larger or more complex backbones (e.g., multi-domain proteins, transmembrane proteins). The absence of cross-domain evaluation weakens the claim that the method enhances "overall designability".

4. The proposed PDB-D dataset consists mainly of AF2-derived per-residue pLDDT labels. Its novelty and added value over existing resources are not clearly demonstrated.

**Questions:**

1. Given that AlphaFold3 now offers more accurate modeling of complexes, ligands, and residue-level confidence, why did the authors choose to rely solely on AlphaFold2 for both preference generation and evaluation?
Would adopting AF3 or combining AF2 and AF3 predictions alter the designability signal or improve validation reliability?

2. Could the authors quantify the additional computational cost of applying DPO at the residue level compared with standard sequence-level DPO, and clarify whether the reported improvements in designability justify this overhead?

3. Do performance gains distribute uniformly across different secondary-structure elements (α-helices, β-sheets, loops), or are improvements concentrated in specific regions?

4. Can the residue-level preference framework (RPL/RCL) be applied to other sequence-design models or diffusion-based models?

---

### Official Review · Reviewer_q5bv · 2025-11-11

**Soundness:** 2
**Presentation:** 2
**Contribution:** 2
**Rating:** 4
**Confidence:** 4

**Summary:**

The paper tackles the gap between sequence recovery and design success (designability) in protein sequence design. It introduces Direct Preference Optimization (DPO) guided by AlphaFold pLDDT as an objective signal to bias generation toward sequences that are more likely to fold to a target backbone. To improve granularity and stability, the authors propose Residue-level DPO (ResiDPO), which applies preference rewards to residues. They fine-tune LigandMPNN with ResiDPO to create EnhancedMPNN, reporting a near 3× increase in in silico design success on an enzyme benchmark.

**Strengths:**

1. The work is well-motivated.  The work targets a well-identified objective mismatch in protein sequence design—optimizing for sequence recovery rather than designability.

2. The presentation is clear and easy to follow.

**Weaknesses:**

1. The methodological novelty appears limited. The work reads primarily as a straightforward application of DPO to protein sequence design.

2. Results should be broken down by secondary-structure class (all-α, all-β, α/β, α+β). All-α targets are typically easier to design than all-β, so aggregate reporting can mask meaningful differences. Please report the fold-class composition of the evaluation sets and stratified metrics to ensure fair comparisons.

3. The paper relies heavily on pLDDT Accuracy for ablations but presents no empirical validation that this proxy tracks the final design success metric. At minimum, provide a correlation/ rank-correlation analysis (and calibration plots) between pLDDT Accuracy and AF2-based success on a smaller held-out subset with rigorous statistical testing.

4. Evidence for data efficiency should be on validation, with overfitting controls. In Section 4.6 and Figure 5, the data-efficiency comparison (ResiDPO vs. DPO with varying sample sizes) should be evaluated on a validation set, not the training set, to support generalization claims.

5. Missing related work. Important DPO-based protein design papers are not cited or discussed in Section 2. Line 141 mentions peptide design. Please include antibody-design applications that use DPO, such as AbDPO and AbNovo. In particular, AbDPO also introduces a residue-level DPO variant and reports effectiveness/efficiency gains over vanilla DPO.

6. Writing and formatting issues requiring proofreading.

   a. Mathematical expressions contain unreadable or stray symbols (e.g., Lines 198, 199, 372).

   b. Line 51: “…in large language models (LLMs) Lla;” — “Lla” appears to be a typo or placeholder; clarify or remove.

**Questions:**

Please address the questions in the Weakness part.

---

### Note · Authors · 2025-12-04

I have read and agree with the venue's withdrawal policy on behalf of myself and my co-authors.